# Baseline Data of Low-Density Polyethylene Continuous Pyrolysis for Liquid Fuel Manufacture

Aleksandr Ketov [1,*], Vladimir Korotaev [1], Natalia Sliusar [1], Vladivir Bosnic [2], Marina Krasnovskikh [3] and Aleksei Gorbunov [4,5]

[1] Department of Environmental Engineering, Perm National Research Polytechnic University, 614990 Perm, Russia; korotaev@pstu.ru (V.K.); nnslyusar@gmail.com (N.S.)

[2] Research Center RTPlast LLC, Nizhnyaya Pervomayskaya Str. 64, 105203 Moscow, Russia; vbosnik@gmail.com

[3] Department of Inorganic Chemistry, Chemical Technology and Technosphere Safety, Perm State National Research University, 614000 Perm, Russia; krasnovskih@yandex.ru

[4] Institute of Technical Chemistry—A Filial of the Perm Federal Research Centre of the Ural Branch of the RAS, 3, Akademik Korolev Str., 614013 Perm, Russia; agorbunof@mail.ru

[5] Department of General Chemistry of Faculty of Soil Science, Agrochemistry, Ecology and Commodity Research, Federal State Budgetary Educational Institution of Higher Education Perm State Agro-Technological University Named after Academician D.N. Pryanishnikov, 23, Petropavlovskaya Str., 614990 Perm, Russia

[*] Correspondence: alexander_ketov@mail.ru; Tel.: +7-902-834-6620

**Abstract:** The recycling of end-of-life plastics is a problem, since small parts can be returned into circulation. The rest is burned, landfilled or recycled into low-quality heating oil by pyrolysis methods. The disadvantages of this method are the need to dispose the formed by-product, pyrolytic carbon, the poor quality of produced liquid fuel and the low productivity of the method associated with the periodicity of the process. In this work, methods of thermogravimetry and chromatography–mass spectrometry (GC-MS) have been used to study the co-pyrolysis products of low-density polyethylene (LDPE) and oxygen-containing substances at the pressures of 4–8 MPa and temperatures of 520–620 °C. Experiments have highlighted the conditions needed for producing of high-quality liquid fuel. Initial data have been prepared for the design of a continuous pyrolysis reactor to dispose polymer waste for the production of bio-oil which would be available to enter the petrochemical products market.

**Keywords:** environmental impact; low-density polyethylene; fuels; pyrolysis; co-pyrolysis; polymers; oils; decomposition





## 1. Introduction

The valorization of polymer wastes, their reuse and the possibility of generating fuels and energy from their disposal represents a more practical and potentially sustainable path to reducing the pressure on landfills and decreasing the environmental impacts associated with these forms of waste products. Modern civilization cannot be imagined without polymers, which surround us everywhere throughout our lives. The annual production of polymers in 2014 reached 300 million tons with an annual growth of 4%. Therefore, reusing end of life polymers is becoming an increasingly urgent task every year.

Despite the fact that combustion for energy remains the main method of plastics recycling, in recent years recycling trends have been actively developed making it possible to obtain various new products from recycled plastics. The interest in the processing of polyolefin polymers is due to the high proportion of such polymers in industrially produced polymers, reaching almost half of the volume. Thus, one review [1] analyzes the main thermochemical pathways in the valorization of polyolefin wastes unsuitable for mechanical processing, for the production of chemicals and fuel.

The production of liquid fuels from polyolefin wastes is also interesting from a marketing point of view because the demand for liquid fuels is consistently high. Among the available processes, pyrolysis and co-pyrolysis are considered to be preferable processes because they allow the direct production of a liquid fraction, similar to commercial gasoline [2,3].

There are various options for the technological design of the pyrolysis process, which make it possible to increase the yield of the target fraction in the form of liquid fuel. For example, it has been proposed to use a mixture of polyolefins [4,5], to make use of catalysts [6] or to add various compounds such as polystyrene [7], cellulose and wood waste [8,9].

Of particular interest is the co-pyrolysis of polyolefins with water. This is due to the fact that water is usually present in the reaction mixture not only as a component of biomass, but also as one of the reaction products. In this case, water at the temperature and pressure in the reactor can be in a supercritical state and exhibit high activity in the ongoing reactions [10]. The high activity of water under similar conditions is confirmed by the aquathermolysis of high molecular weight organic compounds during the process of oil production [11,12].

Based on the foregoing, it can be assumed that pyrolysis to obtain liquid fuel can be a promising trend for solving the problem of recycling secondary polyolefin polymers; however, it is necessary to provide the necessary process conditions to ensure the cracking of polymers to target hydrocarbons.

In our preliminary studies, it has been shown that the required conditions can be provided in an extrusion-type reactor [13]. It was found that the resulting bitumen-like product was formed as a result of the cracking of molecules of the initial polymer, but it did not reach the average molecular weight typical for liquid fuels. Therefore, to solve the problem of the direct synthesis of liquid fuel from polyolefins, it is necessary to investigate the conditions and select substances for co-pyrolysis that provide the maximum yield of the target product.

## 2. Materials and Methods

### 2.1. Feedstock Preparation

The raw material used was secondary LDPE in the form of granules produced at the Bumatica Enterprise (Perm, Russia). For the research, pine sawdust of a fraction less than 0.5 mm, dried at 100 °C, was used.

### 2.2. Pyrolysis Apparatus and Its Operating Procedure

In all the experiments, a steel tubular vertical reactor with a volume of 30 mL was used, the upper part of which contained a pressure gauge and a valve for sampling the gas phase. A test sample weighing 20.0 g was placed into the reactor. To carry out the pyrolysis process, the reactor was placed in an oven heated to the required temperature. After completion of the pyrolysis, the reactor was removed from the oven and cooled to room temperature. A sample of the gas phase was taken through the valve after the complete cooling of the reactor. The liquid fraction was taken from the reactor after completing the process, cooling the reactor and releasing the residual pressure.

### 2.3. Analytical Methods

Thermogravimetric analysis was performed with a STA 449 F1 device for synchronous thermal analysis (NETZSCH-Gerätebau GmbH, Selb, Germany), allowing the thermal analysis of a sample to be performed with a simultaneous recording of its thermal gravimetric and calorimetric characteristics. The heating rate in all the experiments was 20 degrees per minute. The flow of argon was 40 mL/min in the corresponding experiments.

To create an inert atmosphere, high-purity gaseous argon was used (the volume fraction of argon was not less than 99.998%, the volume fraction of oxygen was not more than 0.0002%), and before each experiment, the furnace with the sample was evacuated.

Platinum crucible was used for thermal analysis. Thermocouple was calibrated using reference substances. Baseline correction performed according to the method supplied with the device. The weighed portions of the samples were measured with an accuracy of at least $\pm 1 \cdot 10^{-2}$ mg. The resulting data were processed with an appropriate software NETZSCH Proteus.

GC/MS results were obtained according to the standard analysis for multi compounds organic mixtures. Chromato-mass spectrometric studies were performed using an Agilent 7890B/5977B instrument with an HP-5ms UI column, 30 m × 0.25 mm, using helium as a carrier gas at 1 mL/min, and an electron impact ionization mode (230 °C, 70 eV). Liquid fractions were dissolved in dichloromethane (10–15 mg/mL), the solution was analyzed under the following conditions: a temperature of 40 °C was held for 1 min, at a rate of 4 °C/min it rose up to 300 °C and was held for 1 min, the inlet temperature was 320 °C, the injection volume was 0.2 μL and the split ratio was 29:1. Gas fractions were injected with a gas-tight syringe and analyzed at 35 °C for 3 min; the inlet temperature was 250 °C; the injection volume was 1 μL and the split ratio was 49–59:1. The compounds were identified with the NIST 2017 MS Library Bundle using retention indices.

$^1$H and $^{13}$C NMR spectra were recorded on a Bruker Avance Neo 400 spectrometer using CDCl$_3$ as a solvent. The $^1$H and the $^{13}$C chemical shifts were measured relative to internal HMDSO ($\delta_H$ 0.055 ppm) and solvent signal ($\delta_C$ 77.2 ppm).

*2.4. Calculation Methods*

Within the framework of the task to obtain liquid fuel from LDPE, the pyrolysis product is assumed to consist of many individual substances; therefore, distillation will take place in a wide temperature range, similar to the distillation of natural oil. To determine the qualitative composition of the resulting liquid fuel according to the results of thermogravimetry, it is proposed to determine the fraction of low-boiling hydrocarbons evaporating up to 150 °C in each sample of the obtained liquid fuel according to the TG curve of thermogravimetry in argon. This liquid fraction roughly corresponds to the gasoline fraction, and the proportion of such a low-boiling fraction is hereinafter referred to as $F_L$ (%).

Under the oil refining conditions, the middle fraction, including kerosene and diesel fuel, is distilled at the temperatures of 150–360 °C. In our research, the amount of the middle fraction evaporating in the given temperature range is denoted as $F_M$ (%).

The pyrocarbon remaining after the evaporation of all components is designated as $F_C$ (%) and is calculated as mass of sample after heating up to 600 °C.

Finally, the heavy fraction corresponding to fuel oil is calculated as the difference $F_H = 100 - F_L - F_M - F_C$ (%). Thus, for the quantitative determination of the resulting liquid fuel composition, according to the results of thermogravimetry in argon, the shares of four fractions formed in the process of distillation at atmospheric pressure were determined for each sample: light $F_L$, medium $F_M$, heavy $F_H$ and pyrocarbon $F_C$.

**3. Results and Discussion**

*3.1. Pyrolysis of Pure LDPE*

As a result of the pyrolysis of pure LDPE and LDPE with impurities at temperatures of 510–590 °C and a heat treatment time of 30–60 min, in all cases liquids were obtained that resembled natural oil in appearance and smell. When heated at atmospheric pressure, all the liquids evaporated to one extent or another, which was monitored by the method of thermogravimetry.

For example, Figure 1 shows the results of the thermogravimetric analysis in argon of the LDPE pyrolysis product obtained by heating for 60 min at 590 °C (curve 1). A similar TG curve for LDPE is shown for comparison.

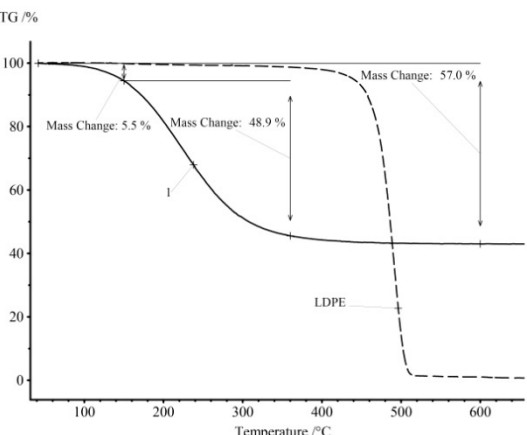

**Figure 1.** The results of the thermogravimetric analysis in an inert atmosphere for the pyrolysis product LDPE (curve 1) and the initial LDPE.

The degradation of mass for LDPE is connected with the chemical process of the decomposition of the polymer structure and carbonization of the material. The pyrolysis product LDPE is a liquid at room temperature, and at the beginning of the heat treatment one can observe the physical process of evaporation of the light fractions, and the pyrolysis process starts only at higher temperatures for heavy fractions. The original LDPE contains practically no light components and undergoes pyrolysis in a narrow temperature range. Unlike LDPE, the resulting pyrolysis fuel starts to evaporate even at room temperature. Thus, the presented sample contains 5.5 mass% of the light fraction $F_L$ and 48.9 wt.% of the middle fraction $F_M$. The amount of pyrocarbon remaining after the distillation of all the hydrocarbons is $F_C = 43.0$ wt.%. The share of the middle fraction is very low—$F_M = 2.6$ wt.%

The optimal heat treatment time and temperature were determined experimentally; for that LDPE was subjected to pyrolysis during various heat treatment periods of time and at various temperatures. The results are presented in Figures 2 and 3.

The number of the target fractions $F_L$ and $F_H$ increases in the pyrolysis liquid, both with increasing the temperature and the heat treatment time. However, the share of the $F_M$ fraction drops sharply with a simultaneous increase in the amount of pyrocarbon $F_C$. This effect becomes unacceptable at high temperatures and longer heat treatment time periods. An increase in the amount of pyrocarbon in the distillation products is accompanied by an increase in the amount of gaseous products, which is confirmed by an increase in the pressure in the reactor. Thus, at 590 °C, the gas pressure was 1.5 MPa in 30 min, and in 60 min of pyrolysis it reached 6.4 MPa. At the same time, at 510°C, the pressure reached only 1.8 MPa in 60 min.

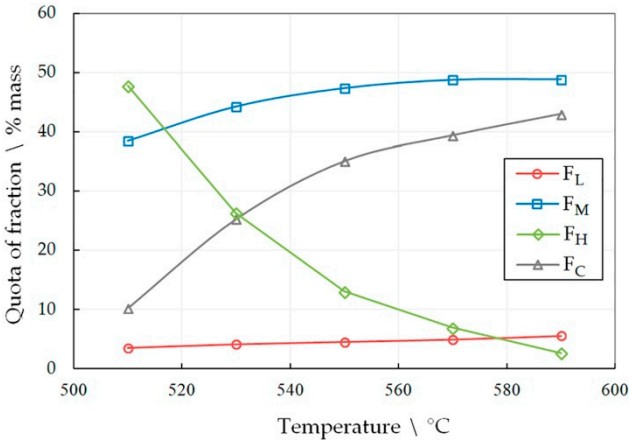

**Figure 2.** The dependence of the share of fractions $F_L$, $F_H$, $F_M$ and $F_C$ in the liquid product of LDPE pyrolysis for 60 min on temperature.

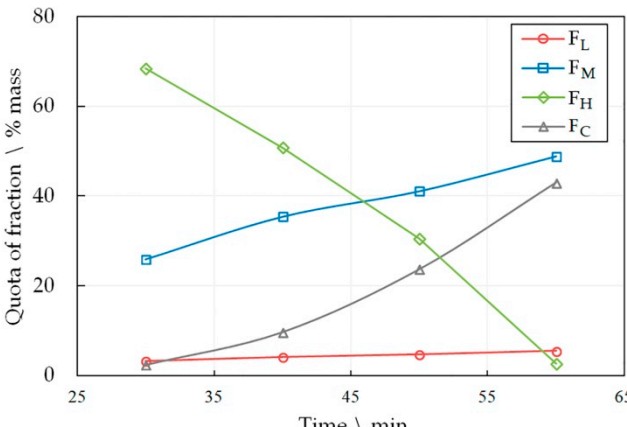

**Figure 3.** The dependence of the share of $F_L$, $F_M$, $F_H$ and $F_C$ fractions in the liquid product pyrolysis of LDPE at 590 °C on the time of the process.

Therefore, a working hypothesis was accepted that, in the process of pyrolysis, the rupture of polymer molecules occurs predominantly near the ends, and predominantly short radicals are formed. The interaction of radicals with each other leads to the formation of light hydrocarbons in the form of gases or relatively light liquids included in the $F_L$ and $F_M$ fractions. At the same time, heavy hydrocarbons $F_H$ are formed in insignificant amounts, and those that are formed split off light radicals from the molecule, being enriched in carbon and increasing the FC fraction. High molecular weight unsaturated hydrocarbons accumulate in the pyrolysis liquid with a corresponding increase in the C/H ratio. As a result, during the distillation of the liquid, polyunsaturated hydrocarbons polymerize and resinify up to the formation of pyrocarbon.

The results of the GC-MS analysis of the obtained liquid are presented in Figure S1. The sample is dominated by n-alkanes, from $C_7$ to $C_{33}$, which make up 32.2%. The proportion of branched alkanes is much less—5.8%, as is that of alkylcyclohexanes and alkylcyclopentanes—9.4%, alkenes—5.4%, unsaturated cycles—2.1% and arenes—34.0%, of which 12.5% belong to PAH.

In this case, the gas phase should have a higher content of hydrogen-saturated substances, starting with methane, not only in comparison with the pyrolysis liquid, but also in relation to the initial LDPE. In the gas products, compounds with low C/H ratios are generally volatile and hence they appear in the gas fraction. In fact, the results of the GC-MS of the gas phase over the pyrolysis liquid demonstrate the predominance of more hydrogen-saturated molecules: saturated over unsaturated ones (51: 7%), and hydrocarbons $C_{1-3}$ over $C_{4-6}$ (51: 49%) (Figure S2).

It can be assumed that in the process of LDPE pyrolysis there is an intense formation of hydrogen and methyl radicals H• and $H_3C$•, which recombine with each other and higher molecular weight radicals. As a result, the C/H ratio decreases in the gas phase and increases in the condensed phase.

The mechanism of the thermal degradation of polyethylene consists of two sorts of scission [14]. One is random scission, and the other is chain-end scission. The random scission of C–C links in polymers causes the molecular weight reduction of a raw polymer increasing the quantity of a liquid phase, and the chain-end scission of C–C links causes the generation of volatile products. Provided the thermal destruction of the polymers proceeds according to this mechanism, the behavior of the polymers during the thermal destruction should be influenced not only by the state of a liquid phase, but also by the state of a gas phase in the reactor, namely the reaction pressure.

From a practical point of view, it is unreasonable to increase the pyrolysis temperature and process time above the identified limits of 590 °C and the heat treatment time of 60 min due to the rapid growth of condensed hydrocarbons in the liquid phase, which cannot be obtained by direct distillation.

Taking into account the concept of the radical nature of the pyrolysis process, one can assume that the addition of substances that supply free radicals to the system can lead to a decrease in condensed hydrocarbons in the product. That is why it is proposed to carry out LDPE pyrolysis in the presence of oxygen-containing compounds.

### 3.2. LDPE Pyrolysis Together with Oxygenated Compounds

Oxygenates were added to LDPE to suppress the formation of condensed co-pyrolysis products on the assumption that oxygenated radicals would facilitate chain termination and prevent double bond formation.

A number of researchers have considered the co-pyrolysis of LDPE with various types of wood waste, which, in fact, is the addition of oxygen-containing polymers in the form of cellulose and lignin as well as residual water in the wood to the reaction mixture.

Thus, the co-pyrolysis of high-density polyethylene (HDPE) and almond shells at 500 °C and atmospheric pressure leads to the production of bio-oils with a high content of carbon and hydrogen, as well as a lower content of oxygen and a higher calorific value than in the case of the bio-oil obtained from almond shells [15]. The elemental analysis results show that the oil obtained by co-pyrolysis is very similar to the currently used transport fuel in terms of H/C ratio and heating (calorific) value. The combined pyrolysis of biological waste with polyethylene reduces the oxygen content in the product, which makes the resulting oil more stable. The $C_{13}$–$C_{25}$ compounds discovered in the aliphatic fraction of the product are characteristic of the diesel fraction of motor fuel. The authors conclude that the addition of high-density polyethylene to biomass in the process of co-pyrolysis at 500 °C and atmospheric pressure has made it possible to obtain biofluids with a yield of about 40–50 wt.%. The authors consider the method of combined pyrolysis of biomass with synthetic polymer to have good prospects for the integrated production of motor fuels and chemicals, taking into account a sufficiently high yield of bio-oil, and the solution of the plastic recycling problem. This conclusion is consistent with the other research data. Mixtures of different types of biomass wood and polyolefinic polymers can be radically converted to liquid products by pyrolysis under an inert atmosphere in autoclave conditions [16]. The feedstock materials used in this work included beech wood, pine wood, cellulose and hydrolytic lignin, as well as commercially available medium density polyethylene, atactic-polypropylene and isotactic-polypropylene.

Co-pyrolysis with wood is possible not only with polyolefins, but also with other polymers. Thus, the co-pyrolysis of styrene-butadiene rubber with lignin-containing materials in the form of alder wood, straw biomass and furniture waste leads to an increase of oil in the pyrolysis products [17]. The mutual influence of pyrolysis processes was shown with polyaromatic polymers by studying the co-pyrolysis of polystyrene with biomass from walnut shells and peach stones using the TGA coupled to FT-IR and MS method [18]. The use of various polymers for co-pyrolysis with lignin, for example, polyethylene, polypropylene, polystyrene and bisphenol, described by the authors [19], does not fundamentally change the picture of the process; there is a positive effect of lignin on the pyrolysis of polymers which allows obtaining bio-oil with a wide range of compounds.

The co-pyrolysis of low-density polyethylene (LDPE) with three types of biomass, cedar wood, sunflower stalk and Fallopia Japonica stem, was carried out in a dropdown tube reactor [20]. It was found that the maximum relative yield of oil in the case of the co-pyrolysis process was obtained at 600 °C, which significantly exceeded the optimal pyrolysis temperature of individual biomass or LDPE. The synergistic effect was positive for the production of aliphatic compounds. In our opinion, these effects are explained by the participation of cellulose molecules as a donor of radicals that initiate the processes of the destruction of the polymer LDPE molecules. This hypothesis makes it possible to explain not only the acceleration of LDPE decomposition (noted by the authors of the article) in the presence of inorganic elements in the form of silica sand and oxides of $K_2O$ and CaO ash, which act as catalysts, but also the pyrolysis scheme proposed by the authors [21],

including the participation of steam, and in this case the catalyst was specially added to the mixture in the form of zeolites. Besides, steam is produced in the reaction zone by the initial reaction components and participates in the process of the chemical transformations of the mixture as an active reagent. For example, specially added catalysts in the form of HZSM-5 and MgO, as well as heat supply from a microwave oven, have a significant effect on the acceleration of the process of the combined pyrolysis of LDPE and lignin at 450–600 °C [22]. The authors also claim the presence of an obvious synergistic effect between lignin and LDPE in terms of the bio-oil yield.

Therefore, consideration of the co-pyrolysis of hydrocarbon polymers with oxygen-containing polymers suggests that the mechanism of the polymer molecules rupture, in this case, differs from the mechanism of the individual polymers' pyrolysis and the difference is possibly associated with the participation of oxygen-containing radicals in the process of the destruction of C–C bonds of the polymer molecules. In this case, the presence of steam in the process of LDPE pyrolysis should promote the formation of low-molecular weight compounds.

The examples described above allow us to suppose that the LDPE pyrolysis process can be initiated not only by wood, or by cellulose and lignin included in its composition, but also by other oxygen-containing polymers and even low-molecular compounds containing oxygen, for example, alcohols and water.

Regarding oxygen-containing compounds, we chose high-molecular compounds for our experiments (starch and palm oil), as well as low-molecular compounds (glycerin, ethyl alcohol and water). Mixtures of LDPE and these compounds were subjected to pyrolysis at 590 °C for 60 min. The resulting liquids were analyzed for the content of the $F_L$, $F_H$ and $F_C$ fractions by the thermogravimetric analysis. The results of the analysis are presented in Table 1. For comparison, the corresponding figures are given for the pyrolysis product of pure LDPE processed under the same conditions. The table also shows the pressure values in the reactor at the end of the pyrolysis process for each mixture.

**Table 1.** Results of the analysis of the pyrolysis liquid obtained from the mixture of LDPE with oxygen-containing substances.

| Composition of the Initial Mixture | Proportion of Impurities in the Initial Mixture, Mass.% | Pressure in the Reactor at the End of Heat Treatment, MPa | $F_L$, Mass.% | $F_H$, Mass.% | $F_C$, Mass.% |
|---|---|---|---|---|---|
| LDPE | | 6.4 | 5.5 | 48.9 | 43.0 |
| LDPE + starch | 15.0 | 7.2 | 13.5 | 47.7 | 10.8 |
| LDPE + palm oil | 15.0 | 6.5 | 2.7 | 39.4 | 17.7 |
| LDPE + glycerin | 15.0 | 7.8 | 11.7 | 63.0 | 11.2 |
| LDPE + ethanol | 15.0 | 7.5 | 4.2 | 56.2 | 14.9 |
| LDPE + water | 7.5 | 7.7 | 3.9 | 53.8 | 18.7 |

An almost complete absence of oxygen-containing compounds in the liquid phase of the samples is confirmed by the NMR data. In the proton spectrum, the 3.5–4 ppm region was practically free of signals; in the $^{13}C$ spectrum, no signals in the range of 45–75 ppm were detected (Figure S3a,b).

Thus, oxygen is concentrated in gaseous products in the form of carbon oxides. The presence of oxygen-containing compounds in the initial mixture leads to a significant increase in the proportion of low-molecular products. This effect is confirmed by the data of the GC-MS analysis of the gases (Table S1) and liquids (Table S2) after pyrolysis of pure polyethylene and polyethylene with the admixture of glycerin.

The ratio of the areas of chromatographic peaks for the co-pyrolysis case to pure LDPE pyrolysis case of individual substances as for gases and as for liquids, proportional to the percentage of the corresponding substance in the mixture, decreases sequentially with the increase in the molecular weight of the studied substances. Obviously, the presence of an oxygen-containing substance in the initial mixture with LDPE leads to a synthesis

of lower molecular weight products. In our opinion, this fact indicates that oxygen-containing substances produce a lot of oxygen-containing radicals during pyrolysis and these radicals actively participate in the process of cracking and creating numerous low-molecular substances. Therefore, for the production of liquid fuel from LDPE, it is preferable to carry out co-pyrolysis with oxygen-containing compounds. The compare of the ratio of various pyrolysis products with the same molecular weight allows us to assume that a greater quota of cycloalkanes and branched isomers are formed in the case of the presence of oxygen-containing substances in the initial mixture compared with pure LDPE. But this assumption needs more experiments and requires further verification.

The results obtained agree with the data of other authors on the radical mechanism of the pyrolysis process and the key role of oxygen-containing radicals in the case of their presence. Thus, certain assumptions about the intermediates that promote the pyrolysis of hydrocarbon polymers can be made on the basis of the work [23]. The authors investigated the process of the co-pyrolysis of polyethylene (LDPE), polycarbonate and polystyrene together with lignin by the thermogravimetry method in a stream of nitrogen. The waste gases were continuously analyzed by the infrared spectroscopy.

Despite the difference of the products identified in the flue gases, what they have in common is the obligatory presence of steam and carbon dioxide in the IR spectra at all temperatures in the case of co-pyrolysis. This suggests that steam and carbon dioxide are indispensable intermediates that promote the pyrolysis of polymers in the bio-oil direction.

The presence of steam as an active reagent contributing to the fragmentation of high molecular weight hydrocarbons is confirmed by the practice of oil production using superheated steam. Simulation of aquathermolysis in a laboratory batch reactor showed an intensive process at temperatures above 240–350 °C and a pressure above 0.5–4.1 MPa [24]. Experimental results show that aquathermolysis does take place for conventional heavy oil. As the reaction time increases, the oil viscosity reduces. However, the reaction will reach equilibrium after a certain period of time and will not be sensitive to any further reaction time anymore. Analysis shows that, while resin and asphaltenes decrease, saturated hydrocarbons and the H/C ratio increase after the reaction. The content of high carbon number hydrocarbons decreases and that of light hydrocarbons increases.

The use of supercritical water in the upgrading of heavy oil feedstock is characterized by high efficiency due to both the transition of the process to more severe conditions with temperatures above 400–500 °C and a pressure above 20–25 MPa, and the changes in the properties of water during the transition to the supercritical state [25]. This process has demonstrated itself as an effective method for upgrading heavy oil feedstock (including tar) to obtain lightweight, high-quality semi-synthetic oil with a minimum coke yield. In our experiments we observed a similar decrease in coking products.

To explain the observed effects, one should consider a possible mechanism of the LDPE pyrolysis process in its pure form and in the presence of oxygen-containing compounds. In general, the pyrolysis of organic compounds always contains free radical mechanisms [26]. The interpretation of such reactions is usually carried out by separation into elementary reactions: forming radicals, consuming radicals and interacting with radicals, which form the building blocks of general mechanisms. The cracking process of alkane polymers is based on the reaction of the homolytic cleavage of molecules at the C–C bond with the formation of radicals. The presence of oxygen heteroatoms in the composition of molecules in the system promotes the formation of various radicals that contribute to the initiation and continuation of chain-type reactions.

The assumption concerning a free radical mechanism was proved by the authors of [27], who showed that a significant amount of high-energy free radicals formed during the pyrolysis of lignin are capable of breaking bonds in hydrocarbon polymers with the formation of many shorter molecules. The predominantly homolytic splitting of molecules at the C–O bond is proved by the method of electron paramagnetic resonance [28] in the case of oxygen-containing organic molecules by the example of the pyrolysis of lignin at 500 °C.

Oxygen-containing radicals obtained in the process of lignin pyrolysis can interact with foreign substances supplied to the system. Thus, it was found by the methods of electronic and magnetic resonance spectroscopy [29] that during the pyrolysis of guaiacol, a monomer of lignin, at 600 °C an o-phenyl semiquinone radical anion, a cyclopentadienyl radical, a hydroxyl radical, a methoxy radical, a phenyl radical and a phenoxy radical were formed.

In addition to the oxygen-containing polymers, alcohols can also be an active supplier of radicals in the system. The direct determination of free radicals by the electron paramagnetic resonance during the pyrolysis of cinnamic alcohol, the simplest non-phenolic lignin model compound at 400–800 °C, is presented by the authors [30]. It is shown that a key role is played by O-centered radicals, which confirms the assumption of the direct participation of oxygen-containing polymer compounds in co-pyrolysis with hydrocarbon polymers.

The research [31], by the method of two-dimensional heteronuclear single-quantum coherence nuclear magnetic resonance combined with a high-temperature oven, showed the multistage nature of the pyrolysis process. The radical reaction of the lignin pyrolysis was divided into three stages, including: the radical inducing stage, the main reacting stage and the quenching stage. In some cases, it was possible to release a significant amount of free radicals, which actively contributed to the decomposition of polymer molecules.

Thus, we can make a conclusion that the addition of oxygen-containing compounds to LDPE during the process of pyrolysis can significantly reduce the proportion of the coking fraction $F_H$ in the liquid pyrolysis products as a result of increasing the proportion of straight-run fractions. This is probably due to the formation at pyrolysis temperatures of a significant amount of short oxygen-containing radicals, which promote the rupture of polymer hydrocarbon molecules and the formation of medium-length hydrocarbons. Active oxygen-containing radicals activate radical chain reactions in hydrocarbons, but they themselves ultimately turn into stable carbon dioxide.

From the data in Table 1, it follows that in the presence of oxygen-containing compounds the proportion of the coking fraction decreases by 2.3–4.0 times, while we cannot speak about an obvious increase in the proportion of $F_L$ and $F_M$ fractions. It is obvious that the decrease in the coking $F_C$ fraction is due to the increase in the $F_H$ fraction. An increase in pressure is observed during the co-pyrolysis of LDPE with oxygen-containing compounds compared with the pyrolysis of pure LDPE, which confirms the hypothesis of the growth of light radicals during the pyrolysis of LDPE in the presence of oxygen-containing radicals.

In the gas phase, in comparison with the pyrolysis of pure LDPE, according to the GC-MS, the proportion of light hydrocarbons increases: the ratio of $C_{1-3}$ to $C_{4-6}$ is 75:23%, and the ratio of saturated ones to unsaturated is 80:18% (Figure S3). At the same time, the proportion of carbon dioxide increases sharply (from 0.1 to 2.5%), which can testify to the transition of all the oxygen from the initial oxygen-containing compounds into carbon dioxide.

## 4. Conclusions

On the basis of the experiments carried out, one can state that at temperatures of 510–590 °C it is possible to carry out the pyrolysis of LDPE during the period of time measured in tens of minutes with the formation of liquid hydrocarbons. The presence of oxygen-containing compounds in the initial mixture contributes to the increase in the liquid products of fractions suitable for direct distillation by an analogy with oil distillation. The possibility of carrying out the process in a continuous flow mode is confirmed by the literature data.

The key problem is the supply of polymers to the reactor. For these purposes, it is possible to use reactive extrusion, when the presence of a feed screw prevents the products from escaping in a countercurrent flow. The solution is applied in practice to carry out various reactions with polymers. For example, the cross-linking of different polymers under reactive extrusion conditions can be used to create biodegradable polymers [32]. On the other hand, reactive extrusion can be used to carry out opposite reactions, for example,

the destruction of polymer molecules. For example, repeated extrusion of a composite material from polypropylene and wood fiber in a twin-screw extruder at a maximum temperature of 205 °C leads to the decrease of the molecular weight of polypropylene [33], which indicates the beginning of the polymer chain degradation reaction. The effect of decreasing the molecular weight is manifested in the TG and DTG curves as a shift of the curves to the low-temperature area, as it was observed in the process described.

Reactive extruders have been shown to be a flexible and useful reactor design for the degradation of plastic [34]. Some examples of reactive extruders can be used for plastic degradation including the thermal and catalytic degradation of polyethylene for the production of liquid fuel, co-processing polyethylene with lubricating oil, and as moving bed reactors that convey both sand and polypropylene for fuel gas production.

The article provides data on the HDPE pyrolysis in an extruder at temperatures up to 425 °C and the duration of the process up to 10–12 min. The authors observed a decrease in the molecular weight of HDPE.

In addition to the cracking of polymer molecules, depolymerization is observed in some cases, as described by the authors [35] for polystyrene. In this case, during the twin screw extrusion, a depolymerization reaction occurs at an ultra-high speed and temperatures up to 280 °C. A model of depolymerization of extruded polystyrene was constructed based on the analysis of the molecular weight ratio of extruded polystyrene depending on the conditions.

Thus, the requirements for the continuous pyrolysis of LDPE with its transformation into a target product in the form of a liquid with a high proportion of straight-run products can be formulated as follows:

- The temperature range of the reaction is near 590 °C.
- The presence time of substances in the reaction zone is from 40 to 60 min.
- The presence of oxygen-containing compounds in the form of impurities in LDPE in the original composition.
- Feeding raw materials to the reaction zone by the extrusion method.
- The reactor must be designed for the pressure of at least 8 MPa in the reaction zone.

Our further efforts will be directed to the creation of a reaction apparatus that satisfies the formulated conditions, with the ultimate aim of creating a continuous process for processing the secondary LDPE into a liquid product with the compact equipment operating in a flow-through mode.

**Supplementary Materials:** The following supporting information can be downloaded at: https://www.mdpi.com/article/10.3390/recycling7010002/s1, Figure S1: Chromatogram of the liquid phase after the pyrolysis of LDPE at 590 °C for 60 minutes, Figure S2: Chromatogram of the gas phase after the pyrolysis of LDPE at 590 °C for 60 minutes, Figure S3: $^1$H (a) and $^{13}$C (b) NMR spectra of the liquid phase, Table S1: Results of analysis of the pyrolysis gases obtained from pure LDPE and the mixture of LDPE with glycerin, Table S2: Results of analysis of the pyrolysis liquids obtained from pure LDPE and the mixture of LDPE with glycerin.

**Author Contributions:** Conceptualization, A.K. and V.B.; methodology, N.S.; software, A.G.; validation, A.K., V.K. and N.S.; formal analysis, M.K.; investigation, M.K.; resources, V.K.; data curation, A.G.; writing—original draft preparation, A.K.; writing—review and editing, A.K.; visualization, M.K.; supervision, V.B.; project administration, N.S.; funding acquisition, V.K. All authors have read and agreed to the published version of the manuscript.

**Funding:** This research was funded by Ministry of science and higher education of the Russian Federation, Project number FSNM-2020-002 "Development of scientific basis for environmentally friendly and nature-inspired technologies and environmental management in petroleum industry".

**Informed Consent Statement:** Not applicable.

**Acknowledgments:** The authors are grateful to V. Galkin for the fruitful discussion and to N. Rossomagina for the technical support.

**Conflicts of Interest:** The authors declare no conflict of interest.

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
