# Peer review of "Baseline Data of Low-Density Polyethylene Continuous Pyrolysis for Liquid Fuel Manufacture"

_recycling, doi:10.3390/recycling7010002_

Round 1

Reviewer 1 Report

            There is probably something publishable here but the paper is so poorly written that it will require major work. The paper appears to be part review and part a study of the effect of adding cellulose to polymers during pyrolysis. Frankly, it is difficult to get much out of this. Specific points are as follows:

  • The English is not terrible but it is stilted. For example, an English speaker would not say “The rest part…” (second sentence of the Abstract). More serious is the fact that the writing is not concise or precise. Sentences are repeated: “To study the pyrolysis of various plastics, …, a tubular reactor was used [4]. The decomposition of plastic waste was carried out in a horizontal tube reactor.”
  • The Introduction is very long and most of it is not relevant. It should not take more than 4 pages to say the paper is an investigation of the effects of co-pyrolysis. Given the current activity in polymer recycling, there is also no excuse for having most of the references be more than 10 years old. Why did the authors highlight 15-year-old paper on the effects of adding ZnO when there are so many recent publications on this? I did a quick search on “polymers” and “pyrolysis” and got more than 3000 hits in the last five years. The third hit, (“Co-pyrolysis of polymer waste and carbon-based matter as an alternative for waste management in the developing world” 1016/j.jaap.2021.105077), seems to be exactly the subject of this paper.
  • How one does pyrolysis will clearly change the results one gets and the description of the authors’ experimental apparatus is not complete. They say it is horizontal tube but what is the tube diameter and volume? Surface area could matter in this kind of system. What was the pressure and how did they control it? If one wanted to reproduce these results, one would need to know more about the polymer and its molecular weight.
  • Showing raw data (chromatograms, etc) is not really helpful. There is probably some useful information in Tables 1 and 2 but there is no real analysis what that data shows.

Author Response

Dear Reviewer!

Thank you for your review and comments. You recommendations are very helpful and have been accepted with gratitude. I would like to respond to your comments point-by-point.

- The English is not terrible but it is stilted. For example, an English speaker would not say “The rest part…” (second sentence of the Abstract).

Accepted. The text has been checked up and corrected by a professional English linguist.

- More serious is the fact that the writing is not concise or precise. Sentences are repeated

Accepted. The repetitions have been eliminated and the text has been shortened.

- The Introduction is very long and most of it is not relevant. It should not take more than 4 pages to say the paper is an investigation of the effects of co-pyrolysis. Given the current activity in polymer recycling, there is also no excuse for having most of the references be more than 10 years old. Why did the authors highlight 15-year-old paper on the effects of adding ZnO when there are so many recent publications on this?

Accepted. The text has been made more than two pages shorter,

- The third hit, (“Co-pyrolysis of polymer waste and carbon-based matter as an alternative for waste management in the developing world” 1016/j.jaap.2021.105077), seems to be exactly the subject of this paper.

Can`t agree with that. The mentioned article [Phakedi, D., Ude, A. U., Oladijo, P. O. Co-pyrolysis of polymer waste and carbon-based matter as an alternative for waste management in the developing world. Journal of Analytical and Applied Pyrolysis. 2021,155, 105077. doi:10.1016/j.jaap.2021.105077] is devoted to the carbon-based waste materials, namely coal. In contrast, our article is about the influence of oxygen-containing substances in co-pyrolysis process with LDPE.

- How one does pyrolysis will clearly change the results one gets and the description of the authors’ experimental apparatus is not complete. They say it is horizontal tube but what is the tube diameter and volume? Surface area could matter in this kind of system. What was the pressure and how did they control it? If one wanted to reproduce these results, one would need to know more about the polymer and its molecular weight.

A “horizontal tube” is not mentioned in our description of the experiments (2.2. Pyrolysis apparatus and its operating procedure), but it is written that the volume of the reactor is 30 ml and the pressure is measured with a pressure gauge. Nevertheless, we added the orientation characteristics of the reactor to be more precise.

The origin of raw LDPE is also mentioned in the article (2.1. Feedstock preparation).

- Showing raw data (chromatograms, etc) is not really helpful. There is probably some useful information in Tables 1 and 2 but there is no real analysis what that data shows.

The corresponding information is given in Table 2.

Reviewer 2 Report

The manuscript is nicely written and I am sure results will find direct applications in the widely discussed waste to energy approach. Below are my minor comments:

Please describe the chemical events in the TGA profile in Figure 1.

How obtained GC/MS results compared with analogous studies from the literature?

In the gas products, compounds with low C/H ratios are generally volatile and hence they appear in the gas fraction. 

I would add a scheme that describes chemical reactions that prevail in the pyrolytic medium.

Is there a significant effect of pressure? would distribution of products will be different if experiments were carried out at 1 atm?

In the introduction, I would add a note that a large fraction of plastic are actually contaminted with halogens as in BFRs. 

Author Response

Dear Review!

Thank you for your review and comments. You recommendations have been very helpful and accepted with gratitude.

I would like to respond to your comments point-by-point .

- Please describe the chemical events in the TGA profile in Figure 1.

Accepted (the text below Figure 1 )

- How obtained GC/MS results compared with analogous studies from the literature?

Accepted (the text below Section 2.3. )

- In the gas products, compounds with low C/H ratios are generally volatile and hence they appear in the gas fraction.

Accepted (the text below Figure 4 has been added)

- I would add a scheme that describes chemical reactions that prevail in the pyrolytic medium.

The scheme that describes chemical reactions that prevail in the pyrolytic medium according to the authors [Murata, K., Sato, K., Sakata, Y. Effect of pressure on thermal degradation of polyethylene. Journal of Analytical and Applied Pyrolysis. 2004. 71 (2), 569-589.  https://doi.org/10.1016/j.jaap.2003.08.010   ] has been added to the description.

- Is there a significant effect of pressure? would distribution of products will be different if experiments were carried out at 1 atm?

The description of pressure influence has been  added to the text.

- In the introduction, I would add a note that a large fraction of plastic are actually contaminted with halogens as in BFRs.

The problem of halogens presence in waste plastics is great and difficult. Even without its discussion the article turned out to be rather long. That is why we avoided discussing this subject and even mentioning it.